# On-Site and Visual Detection of the H5 Subtype Avian Influenza Virus Based on RT-RPA and CRISPR/Cas12a

**DOI:** 10.3390/v16050753

**Published:** 2024-05-10

**Authors:** Xu Zhou, Siwen Wang, Yue Ma, Yongping Jiang, Yanbing Li, Jianzhong Shi, Guohua Deng, Guobin Tian, Huihui Kong, Xiurong Wang

**Affiliations:** State Key Laboratory of Animal Disease Control and Prevention, Harbin Veterinary Research Institute, Chinese Academy of Agricultural Sciences, Harbin 150069, China; 82101215654@caas.cn (X.Z.); siwen00000@163.com (S.W.); 15188228175@163.com (Y.M.); jiangyongping@caas.cn (Y.J.); liyanbing@caas.cn (Y.L.); shijianzhong@caas.cn (J.S.); dengguohua01@caas.cn (G.D.); tianguobin@caas.cn (G.T.)

**Keywords:** CRISPR/Cas12a, RT-RPA, H5 AIVs, visual detection

## Abstract

Avian influenza viruses (AIVs) of the H5 subtype rank among the most serious pathogens, leading to significant economic losses in the global poultry industry and posing risks to human health. Therefore, rapid and accurate virus detection is crucial for the prevention and control of H5 AIVs. In this study, we established a novel detection method for H5 viruses by utilizing the precision of CRISPR/Cas12a and the efficiency of RT-RPA technologies. This assay facilitates the direct visualization of detection results through blue light and lateral flow strips, accurately identifying H5 viruses with high specificity and without cross-reactivity against other AIV subtypes, NDV, IBV, and IBDV. With detection thresholds of 1.9 copies/μL (blue light) and 1.9 × 10^3^ copies/μL (lateral flow strips), our method not only competes with but also slightly surpasses RT-qPCR, demonstrating an 80.70% positive detection rate across 81 clinical samples. The RT-RPA/CRISPR-based detection method is characterized by high sensitivity, specificity, and independence from specialized equipment. The immediate field applicability of the RT-RPA/CRISPR approach underscores its importance as an effective tool for the early detection and management of outbreaks caused by the H5 subtype of AIVs.

## 1. Introduction

Avian influenza viruses (AIVs) are categorized into sixteen hemagglutinin (HA) and nine neuraminidase (NA) subtypes, classifications determined by the antigenic properties of their surface glycoproteins [1,2,3]. These viruses are further divided based on their pathogenicity into highly pathogenic AIV (HPAIV) and low pathogenic AIV (LPAIV), with the H5 subtype being notably one of the few capable of evolving into HPAIV [4]. The H5 subtype AIV is the most frequently detected strain and has caused numerous outbreaks in poultry and wild birds in many countries [5,6]. Alarmingly, these viruses also possess the zoonotic potential to infect humans, leading to severe morbidity and mortality [7]. In light of these challenges, the development of a rapid, sensitive, and easily deployable detection method for H5 viruses becomes critically important for their effective surveillance and management.

Traditional methods for detecting H5 viruses, including virus isolation, serological test, reverse transcription polymerase chain reaction (RT-PCR) or reverse transcription-quantitative PCR (RT-qPCR) [8], and loop-mediated isothermal amplification (LAMP) [9], are known for their time-consuming processes, labor-intensive nature, risk of false positives, and reliance on highly skilled personnel and advanced equipment. Although RT-PCR and RT-qPCR are favored for their high specificity and sensitivity, particularly RT-qPCR with its virus sequence-specific fluorescent probe, their widespread use is limited by the need for sophisticated laboratories and well-trained operators, making them less feasible for rapid and on-site diagnosis in regions with limited resources.

Recent research underscores the potential of the CRISPR-Cas system, the clustered regularly interspaced short palindromic repeat sequences and associated nucleases system, as cutting-edge point-of-care testing (POCT) platforms [10]. These systems are applied to detect a wide array of targets, including viruses, bacteria, parasites, cancer mutations, various genotypes, and small molecules, earning recognition as the forefront of detection technologies [11,12,13,14,15,16,17,18]. Guided by CRISPR RNA (crRNA), the Cas12a nuclease–crRNA complex identifies and cleaves specific DNA sequences, triggering Cas nucleases to indiscriminately cut nearby non-target single-stranded DNA, which is labeled with fluorophore and quencher or biotin [19,20]. This action is reported through fluorescence, blue light, and lateral flow assays (Figure 1) [18]. At present, the Cas nucleases most frequently utilized in CRISPR-based detection are Cas12 (also known as Cpf1), Cas13 (C2c2), and Cas14 from Class II [21,22]. Despite their widespread application, the analytical sensitivity of these CRISPR/Cas systems falls below the thresholds required for clinical diagnostics, necessitating the incorporation of pre-amplification methods like (recombinase polymerase amplification) RPA to augment their sensitivity [23,24]. Recently, various innovative diagnostic platforms have emerged, including Cas13a-SHERLOCK (Specific High-sensitivity Enzymatic Reporter Unlocking) [25], multiple SHERLOCK [26], Cas12a-DETECTR (DNA Endonuclease-targeted CRISPR Trans Reporter) [27], Cas12a-HOLMES (One-hour Low-cost Multipurpose Highly Efficient System) and Cas14-DETECTR [28]. Among these, the CRISPR/Cas12a system stands out for its successful application in identifying pathogens like SARS-CoV-2 [29,30,31], monkeypox virus [32,33,34], HPV16/ 18, and African swine fever virus [35,36], showcasing its pivotal role in advancing pathogen detection.

To date, our review of the literature indicates an absence of any CRISPR/Cas12a-based platforms specifically designed for the detection of H5 subtype AIVs. Addressing this critical gap, our study established an innovative detection system that merges the precision of CRISPR/Cas12a with the rapidity of RT-RPA and the simplicity of visual detection. This method stands to revolutionize on-site detection of H5 subtype AIVs, facilitating timely and effective disease management and control strategies.

## 2. Materials and Methods

### 2.1. Materials

The HA gene in the pMD18T vector was derived from the vaccine seed virus A/chicken/Liaoning/SD007/2017(H5N1). The representatives of 16 subtypes of AIVs (H1-H16), Newcastle Disease Virus (NDV), Infectious Bronchitis Virus (IBV) and Infectious Bursal Disease Virus (IBDV), along with 81 clinical swab samples of H5 AIVs, were isolated and identified nationwide by the National Avian Influenza Reference Laboratory, Harbin Veterinary Research Institute of Chinese Academy of Agricultural Sciences and preserved in the ultra-low-temperature refrigerator. And these clinical swab samples were isolated from poultry farms in 2023.

### 2.2. Reagents and Instruments

RNA extraction kit was purchased from TIANGEN Biotechnology Co., Ltd. (Beijing, China). RT-RPA kit was obtained from Genenode Biological Co., Ltd. (Wuhan, China). Lateral flow strips were purchased from Wobo Biotechnology Co., Ltd. (Nanjing, China). RNase inhibitor was obtained from Thermo Fisher (USA) Co., Ltd. (Nanjing, China). The RT-qPCR reaction reagents were purchased from GuanMu Biotechnology (Changsha, China). RNA purification kit was obtained from Sangong Biotechnology Co., Ltd. (Shanghai, China). T7 transcription kit purchased from New England Biolabs (Ipswich, MA, USA). LbCas12a protein was purified based on protocols established in our team’s previous research. Additionally, the primers and probes in this study were synthesized by Rui Biotech (Beijing, China).

### 2.3. Preparation of crRNA and RNA Template

For crRNA preparation, complementary ssDNA templates containing a T7 promoter of crRNA were used to generate dsDNA by annealing in an annealing buffer. For RNA template production, the target sequences of H5 HA were amplified using primers containing a T7 promoter. The products were identified through gel electrophoresis and purified via Gel Extraction Kit (Omega, GA, USA). crRNAs and target RNA were transcribed overnight at 37 °C using the above products according to the instructions of the T7 RNA Synthesis Kit. Both crRNAs and target RNA were purified with an RNA Purification Kit and either used in subsequent experiments or stored at −80 °C for future use.

### 2.4. RT-RPA Reaction

RT-RPA was performed by a commercial kit (Wuhan, China) according to the principles of the instructions. Briefly, a 50 μL reaction containing 29.4 μL A buffer, 12.1 μL RNase-free H_2_O, 2 μL RNA template, 2 μL forward primer (10 μM), and 2 μL reverse primer (10 μM), 0.5 μL RNase H and 2.5 μL B buffer was mixed and incubated at 42 °C for 30 min. Subsequently, the RT-RPA products were transferred to the CRISPR/Cas12a reaction system.

### 2.5. CRISPR/Cas12a Detection Reactions

For the fluorescence detection, the reaction mix of 25 μL was prepared, incorporating 12.5 μL buffer (20 mM Tris-HCl pH 7.5, 1 mM DTT, 5 mM MgCl_2_, 100 mM KCl, 5% glycerol), 1.25 μL LbCas12a protein, 1 μL crRNA (200 nM), 1 μL ssDNA probe (10 nM), 0.25 μL RNase inhibitor, 2–4 μL RT-RPA products or plasmids, and RNase-free H_2_O to adjust the volume. The reaction was then incubated in a real-time PCR detection System (ABI, USA) at 37 °C for 30–120 min, with fluorescence readings taken every 5 min.

The lateral flow strip assay was conducted using a reaction mixture totaling 50 μL, comprising 25 μL buffer, 2.5 μL LbCas12a protein, 2 μL crRNA (200 nM), 2 μL ssDNA probe (1 μM), 0.5 μL RNase inhibitor, 4 μL RT-RPA products or plasmids, and 14 μL RNase-free H_2_O. This mixture was incubated at 37 °C for a duration ranging from 30 to 120 min. Then the test strip was inserted into the reaction tube, and the result was observed for 2–5 min. A positive result is indicated by the presence of a band at the test line or on both the control and test lines. Conversely, the presence of a band solely on the control line signifies a negative result. The absence of bands on either line suggests an invalid result, necessitating a repetition of the assay.

### 2.6. Design and Screening for crRNA and Primers of RT-RPA

In this study, a total of 988 sequences of the HA gene of Chinese H5 subtype viruses were retrieved from the NCBI database. Utilizing the MegAlign software (v11.0.13) for sequence comparison and analysis, we pinpointed conserved regions within these sequences. Based on the property of LbCas12a to recognize T-rich PAM sequences, six crRNAs targeting distinct conserved regions of the H5 HA gene were designed. The crRNAs demonstrating the most effective results were selected for further experiments (Table 1). Following the selection of the most effective crRNAs, RT-RPA primers targeting the HA gene were designed (Table 2). These primers were then subjected to a comprehensive screening process, where various combinations were tested to identify those with the highest amplification efficiency for use in further experiments.

### 2.7. RT-qPCR Assay for H5 Subtypes AIV

The reaction system for RT-qPCR consisted of 12.5 µL buffer, 1.25 µL RT-enhancer, 0.25 µL Verso Enzyme, 0.8 µL forward and reverse primers (10 µM), 0.6 µL probe (10 µM), and 4 µL of RNA template, with the volume completed to 25 µL with RNase-free water. The reaction program using the QuantStudio 5 (Thermo Fisher Scientific, Waltham, MA, USA) system was reverse transcription at 50 °C for 15 min, pre-denaturation at 95 °C for 15 min, 40 cycles of denaturation at 95 °C for 15 s, and extension at 60 °C for 1 min, and the fluorescence signal was collected at the end of each extension step.

### 2.8. Statistical Analysis

The experiment data were analyzed using GraphPad Prism 8.0. Data are expressed as mean ± standard deviation of three independent experiments.

## 3. Results

### 3.1. Screening of crRNA and Primers

For the CRISPR/Cas12a detection, the target gene was first amplified by RT-RPA. Subsequently, the target DNA was recognized by specific crRNA, forming a ternary complex with Cas12a. This complex enables Cas12a to cis-cleave the target DNA, while also allowing Cas12a to indiscriminately trans-cleave the ssDNA probes (5′-FAM-TTATT-3′-BHQ1-labeled) in the reaction system, thereby releasing fluorescence. To identify the most effective crRNA, six candidates, listed in Table 1, were screened over two rounds of experiments. First, the CRISPR/Cas12a reaction using HA plasmid at a concentration of 9.57 × 10^10^ copies/µL as a template was used to evaluate the efficacy of six crRNAs. In the first round of testing, crRNA-1, crRNA-5, and crRNA-6, which exhibited the higher fluorescence value, were considered the most effective. Subsequently, a second set of evaluations was performed using 10-fold serial dilutions of the plasmid templates to further evaluate the crRNAs. The subsequent tests demonstrated that crRNA-5 had the best performance at low template concentrations; therefore, crRNA-5 was used for all subsequent experiments (Figure 2).

Since there is still no software to design primers for RPA, the amplification of the primers is critical to the detection efficiency. Then, we designed three pairs of primers to amplify the HA gene of H5 viruses after analyzing 988 HA sequences from Chinese H5 viruses. The RT-RPA assay was performed with RNA templates, which were extracted from a H5 virus A/Duck/GuangXi/1/2018 (H5N6), at a concentration of 60 ng/µL by a random combination of three forward primers and three reverse primers, respectively, and then the products were detected for CRISPR/Cas12a, and the pair of primers (F3/R1) with the highest fluorescence value was used for the subsequent assay (Figure 3).

### 3.2. Specificity Test of the RT-RPA/CRISPR

To evaluate the specificity of the established assay, a comprehensive panel of nucleic acids was tested, including all the H1-H16 subtypes of AIV, NDV, IBV, and IBDV. The selected primer pair, F3/R1, together with crRNA5, were used for this rigorous evaluation. Only H5 subtype AIV achieved noticeable fluorescence values (Figure 4A,D), showed color under blue light (Figure 4B,E), and a band on the Test line of the lateral flow strip (Figure 4C,F), which were not observed for the other subtypes of AIV, NDV, IBV, and IBDV. The findings indicated that this assay was uniquely selective for the H5 subtype of AIVs, and had no cross-reactivity with other AIV subtypes or the aforementioned avian viruses. The results demonstrated the exceptional specificity of our approach.

### 3.3. Sensitivity Test of the RT-RPA/CRISPR

In crRNA screening experiments (Figure 2B), it was able to conclude that only CRISPR/Cas12a detection reached a sensitivity of 9.57 × 10^8^ copies/µL without RT-RPA, a level deemed insufficient for the stringent requirements of molecular diagnostics. To further probe the analytical sensitivity of the combined RT-RPA/CRISPR assay, RNA from the H5 HA gene was serially diluted in tenfold increments, ranging from 1.9 × 10^11^ copies/µL down to 1.9 copies/µL, serving as templates. The enhanced sensitivity of the RT-RPA/CRISPR assay was observed to reach as low as 1.9 copies/µL when analyzed through fluorescence values (Figure 5A,B), blue light visualization (Figure 5C), and 1.9 × 10^3^ copies/µL with lateral flow (Figure 5D). All of the above results underscore the assay’s capability for precise and reliable clinical detection of H5 subtype AIVs (Figure 5).

### 3.4. Clinical Samples Detection by the RT-RPA/CRISPR and RT-qPCR

To validate the reliability of the RT-RPA/CRISPR in the detection of H5 subtype AIVs from clinical samples, a comprehensive analysis was conducted. This analysis included 57 swab samples previously confirmed as positive and 24 samples verified as negative through virus isolation. Concurrently, the above samples were also tested by RT-qPCR for comparison. The results, which were interpreted by fluorescence value (Figure 6A), color display (Figure 6B), and lateral flow strip (Figure 6C), indicated that the RT-RPA/CRISPR assay had a positive detection rate of 80.70%, slightly higher than RT-qPCR’s 78.95% (Appendix A), with both methods having a negative detection rate of 87.50% (Table 3). This comparative analysis reveals the RT-RPA/CRISPR assay’s slightly superior detection capability relative to RT-qPCR. Collectively, these results highlight the considerable promise of RT-RPA/CRISPR for the on-site detection of H5 subtype AIVs, eliminating the dependency on advanced and expensive laboratory equipment.

## 4. Discussion

H5 subtype AIVs represent the most critical threat to the poultry industry, characterized by their capacity to cause substantial morbidity and mortality rates among poultry populations. Moreover, these viruses harbor the potential to cross species barriers, posing a significant risk of zoonotic transmission to humans [37]. Influenza viruses bearing the H5 HA have been spreading among wild birds and domestic poultry for more than six decades since the first recorded detection in chickens in Scotland in 1959 [38], causing enormous economic losses and compromising food security. In addition, from 2003 to September 2023, the H5 viruses have caused 965 infections and 491 deaths worldwide. This underscores the urgent need for quick, precise, and reliable diagnostic tools for H5 subtype AIVs, which are vital for preventing their spread and safeguarding public health.

Selecting the optimal crRNAs and primers is key to increasing the efficiency of the CRISPR/Cas12a and RT-RPA assays. Given the propensity of H5 viruses to undergo frequent mutations and the great variability in the nucleotide sequences of the HA gene, which can lead to off-target effects and mismatches, an analysis of 988 sequences of the Chinese H5 HA gene from the NCBI database was conducted to pinpoint a conserved region. This analysis led to the design and synthesis of six crRNAs targeting distinct conserved areas within the HA gene. Through rigorous screening, the crRNA demonstrating the most effective targeting capability was selected. On this basis, three forward and reverse primers were designed, and the primer pair with the optimal amplification effect was identified through a random combination of primers. In addition, the ssDNA reporter is also crucial for nucleic acids detection in the CRISPR/Cas12a reaction, significantly influencing the analytical sensitivity and how detection results are visualized. Previous studies indicated that the use of hairpin ssDNA reporters enhances the trans-cleavage activity of CRISPR/Cas12a [39]. Additionally, the reaction buffer used in this study has been proven to have favorable effects on CRISPR/Cas12 reactions in our previous work, further optimizing the assay’s performance.

The analytical sensitivity of the RT-RPA/CRISPR assay underscores its capacity for the effective detection of H5 viruses. Without pre-amplification, the fluorescence-based Cas12a detection system yielded a sensitivity of 9.57 × 10^8^ copies/μL. However, the Limit of Detection (LoD) for the RT-RPA/CRISPR system was significantly enhanced, achieving levels as low as 1.9 copies/μL when visualized by blue light, and 1.9 × 10^3^ copies/μL using lateral flow strips. Notably, this method showed no cross-reactivity with other subtypes of AIV, NDV, IBV, and IBDV, proving its distinguished specificity in clinical testing. The detection procedure of this method can be finished within about 1 h. To evaluate the clinical applicability of the RT-RPA/CRISPR system, it was applied to 57 clinically derived H5-positive swab samples and 24 H5-negative samples, as confirmed by virus isolation. The comparison of coincidence rates between this method and RT-qPCR against virus isolation outcomes indicated positive coincidence rates of 80.70% for RT-RPA/CRISPR and 78.95% for RT-qPCR, with both methods exhibiting negative coincidence rates of 87.50%. Although there existed false-positive data (Appendix A), these findings highlight the RT-RPA/CRISPR system’s robust performance and its potential for the rapid clinical detection of H5 viruses.

To date, isothermal amplification integrated with CRISPR/Cas12a-based nucleic acids tests has been used for various pathogen detection. However, there are still some shortcomings that need further improvement. The high sensitivity associated with RT-RPA/CRISPR techniques, while beneficial for detection capabilities, introduces the challenge of managing aerosol contamination, a concern that mandates strict laboratory practices. Moreover, the propensity for off-target interactions and mismatches, alongside the requirement for T-rich PAM sequences by Cas12a, highlights the need for improved accuracy in these systems. Also, the target concentration is close to the LOD level of this assay, which makes it challenging to determine the readout accurately, especially when using the lateral flow strips. Financial considerations also emerge as a barrier, with the per-reaction cost of RT-RPA/CRISPR surpassing conventional RT-PCR and RT-qPCR methods. Therefore, future efforts should aim to streamline the detection process, reducing costs to extend the applicability of these assays from clinical laboratories to point-of-care and even home-based testing scenarios.

To date, various assays for H5 subtype AIVs have been established based on different detection principles. For example, Wen et al. established a triple real-time PCR assay for the detection of H3, H4, and H5 subtypes of AIV with a detection threshold of 2.1 × 10^2^ copies/µL [40]. Li et al. created two H5 subtype AIVs assays using real-time fluorescence reverse transcription recombinase-aided amplification (RF-RT-RAA) and RT-RAA coupled with lateral flow dipstick (RT-RAA-LFD), the lowest detectable limits were 1 copies/µL [41]. And they also developed a H5 subtype AIVs assay based on CRISPR/Cas13a, RT-RAA, and LFD with a sensitivity of up to 0.1 copies [42]. In addition, Sączyńska V et al. established a novel epitope-blocking ELISA (EB-ELISA) for the specific and sensitive detection of anti-HA antibodies against the H5 subtype [43]. The sensitivity and specificity of the assay established in this study are comparable to those described above, but the method has advantages in terms of diversified forms of presentation, time-saving, and not strictly relying on specialized instruments.

In this study, we have successfully developed and validated the RT-RPA/CRISPR assay, demonstrating its efficacy for the specific, sensitive, portable, and rapid detection of H5 subtype AIVs. This innovative approach not only offers substantial improvements over traditional diagnostic methods in terms of speed and sensitivity but also introduces a portable solution capable of on-site application. The potential of RT-RPA/CRISPR for immediate deployment in field settings underscores its value as a powerful instrument in the early surveillance and control of H5 subtype AIV outbreaks. By enabling early detection, this assay facilitates timely intervention strategies to mitigate the spread of the H5 virus, thereby contributing significantly to public health and biosecurity.

## Figures and Tables

**Figure 1 viruses-16-00753-f001:**
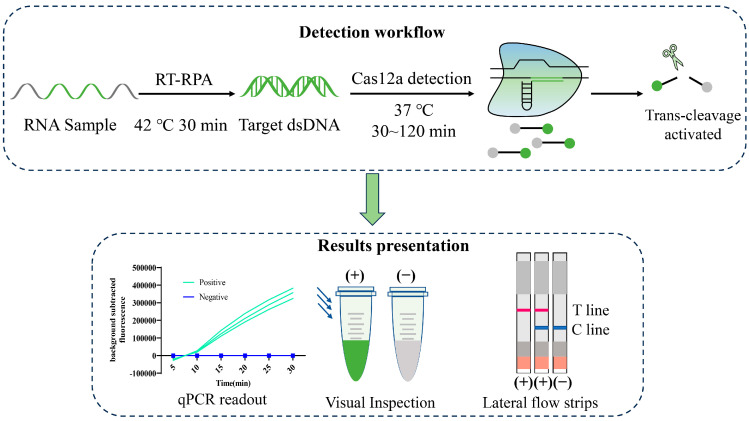
Principles of detection utilizing reverse transcription-recombinase polymerase amplification (RT-RPA) and CRISPR/Cas12a. In this detection system, the target gene fragment from extracted nucleic acids was amplified by RT-RPA. Then, the target DNA was recognized by the specific crRNA and formed a ternary complex with Cas12a, which cis-cleaves the target DNA, and the Cas12a indiscriminately trans-cleaves the ssDNA probes (5′-FAM-TTATT-3′-BHQ1-labeled) or (5′-FAM-TTATT-3′-Biotin-labeled) in the reaction system. Detection results are presented in three ways: fluorescence values, blue light, and lateral flow strips. Interpretation of results of lateral flow strips: A positive result is indicated by the presence of a band at the test line or on both the control and test lines. Conversely, the presence of a band solely on the control line signifies a negative result. The absence of bands on either line suggests an invalid result, necessitating a repetition of the assay.

**Figure 2 viruses-16-00753-f002:**
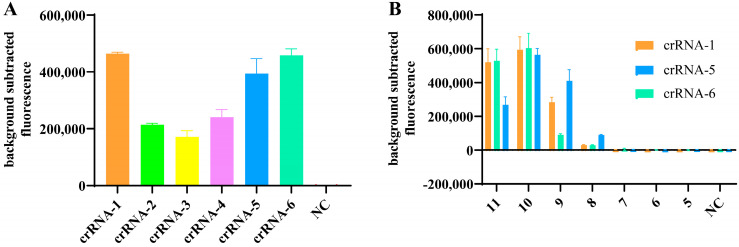
Screening of crRNAs. (**A**) Cleavage activity of CRISPR/Cas12a guided by six crRNAs targeting the HA gene of H5 viruses. (**B**) CRISPR/Cas12a reactions were performed with 10-fold serial dilutions of plasmid templates, and crRNAs with the highest fluorescence values after screening from crRNA-1, crRNA-5, and crRNA-6 were used for subsequent experiments. Fluorescence signals were read using QuantStudio software (Applied Biosystems, v1.4.3). (11~5: 9.57 × 10^11^ copies/µL~9.57 × 10^5^ copies/µL.)

**Figure 3 viruses-16-00753-f003:**
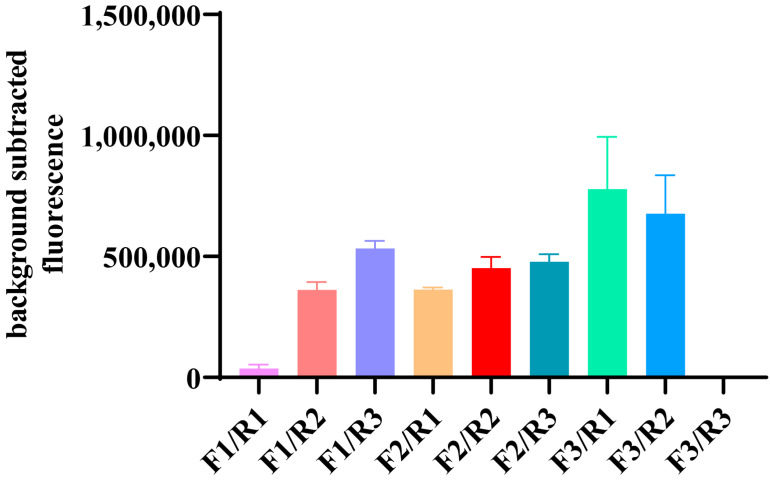
Primer screening. The primer pair with the best amplification efficiency was selected for subsequent experiments by random combining of three forward primers and three reverse primers. Fluorescence signals were read using QuantStudio software (Applied Biosystems, v1.4.3).

**Figure 4 viruses-16-00753-f004:**
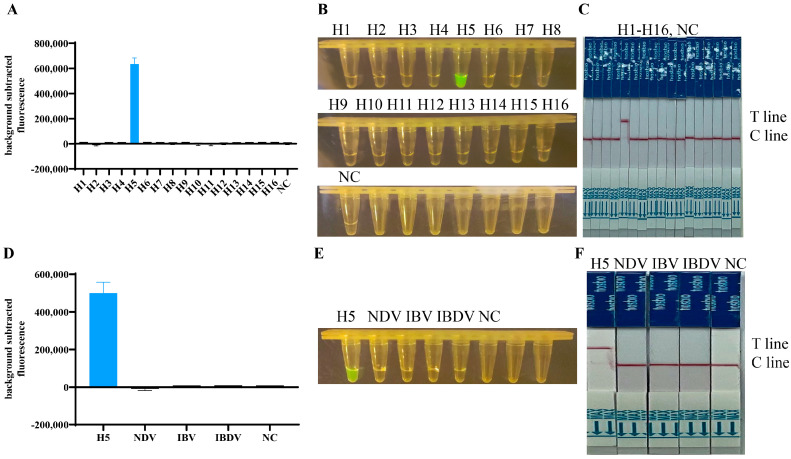
Specificity analysis. (**A**–**C**) RNAs of H5 subtypes and other H1–H16 subtypes of AIV were extracted from inactivated AIVs and used as templates, the HA genes were targeted for RT-RPA/CRISPR specificity assays. (**D**–**F**) RNAs of H5 subtypes, NDV, IBV, and IBDV were used as templates, the HA genes were targeted for RT-RPA/CRISPR specificity assays. The results were presented in three ways: fluorescence value, blue light, and lateral flow strips. Fluorescence signals were read using QuantStudio software (Applied Biosystems, v1.4.3).

**Figure 5 viruses-16-00753-f005:**
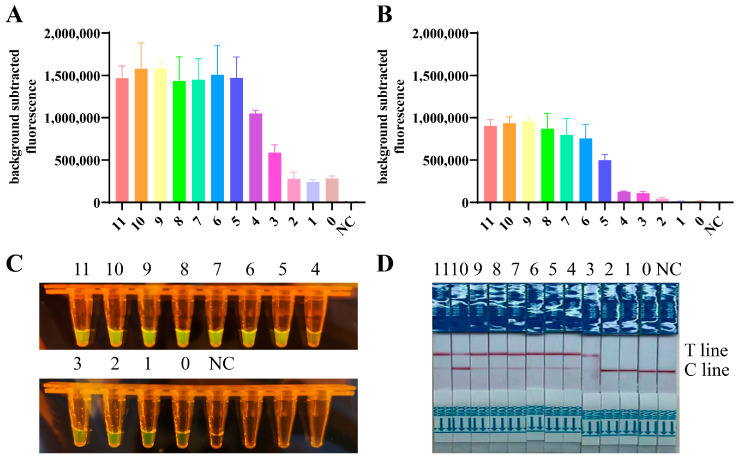
Sensitivity analysis. Tenfold series of dilutions of RNA targeting the HA gene at 1.9 × 10^11^ copies/µL for RT-RPA/CRISPR sensitivity assays. (**A**,**B**) Fluorescence values of RT-RPA/CRISPR targeting H5 HA RNA templates were detected at 2 h and 30 min, respectively. (**C**,**D**) Blue light and lateral flow strips of RT-RPA/CRISPR targeting H5 HA RNA templates; fluorescence signals were read using QuantStudio software (Applied Biosystems, v1.4.3). (11–0, 1.9 × 10^11^ copies/µL–1.9 × 10^0^ copies/µL.)

**Figure 6 viruses-16-00753-f006:**
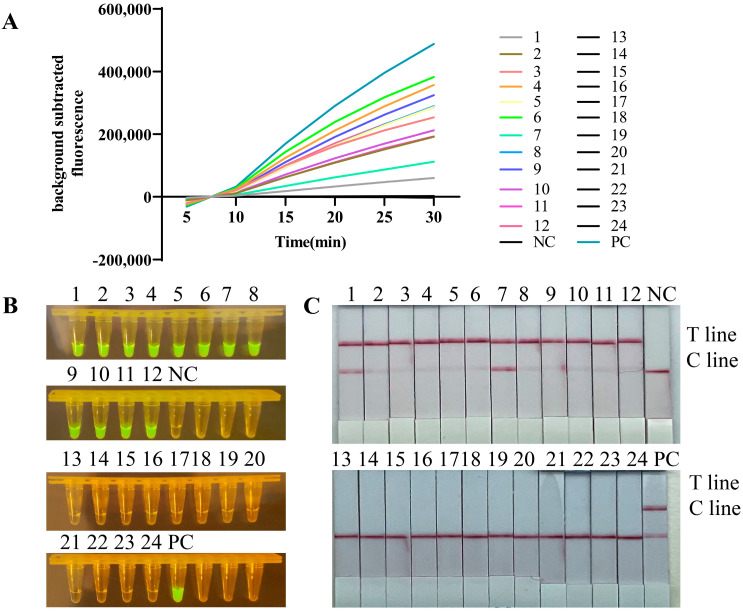
Detection of clinical samples by RT-RPA/CRISPR via (**A**) fluorescence value, (**B**) color display, and (**C**) lateral flow strip. The results of RT-RPA/CRISPR assays performed on 12 positive swabs and 12 negative swab samples were listed randomly. 1–12 were positive samples and 13–24 were negative samples. PC: Positive Control; NC: Negative Control. Fluorescence signals were read using QuantStudio software (Applied Biosystems, v1.4.3).

**Table 1 viruses-16-00753-t001:** The nucleotide sequences of crRNA and probes.

Name	Sequence (5′→3′)
crRNA-1	UAAUUUCUACUAAGUGUAGAUCAUUGGUUACCAUGCAAACAA
crRNA-2	UAAUUUCUACUAAGUGUAGAUCAUGGUAACCAAUGCAAAUCU
crRNA-3	UAAUUUCUACUAAGUGUAGAUCUUGGGAUCUAGUAGCUAUUU
crRNA-4	UAAUUUCUACUAAGUGUAGAUAACAAGAAAAUGGAAGACGGA
crRNA-5	UAAUUUCUACUAAGUGUAGAUCAUGAUUCAAAUGUCAAGAACC
crRNA-6	UAAUUUCUACUAAGUGUAGAUGUAAGUUCCUAUUGAUUCCAA
fluorescence probe	6-FAM-TTATT-BHQ1
strip probe	6-FAM-TTATT-Biotin

**Table 2 viruses-16-00753-t002:** The nucleotide sequences of primers of RT-RPA.

Name	Sequence (5′→3′)
RT-RPA-F1	CTGAACTTCTAGTTCTCATGGAAAACGAGAGG
RT-RPA-F2	AAAATGGAAGACGGATTCCTAGATGTCTGGACC
RT-RPA-F3	CTGGACCTATAATGCTGAACTTCTAGTTCTCATGG
RT-RPA-R1	GAAACAGCCGTTACCCAGCTCCTTTGCATTATCC
RT-RPA-R2	TGTGATAGAATTCGAAACAGCCGTTACCCAGCTCC
RT-RPA-R3	CATTATCGCATTTGTGATAGAATTCGAAACAGCCG

**Table 3 viruses-16-00753-t003:** The detection results of RT-RPA/CRISPR and RT-qPCR assay in clinical samples.

Methods	RT-RPA/CRISPR		RT-qPCR	
Judge	Positive	Negative	Positive	Negative
Positive	46	3	45	3
Negative	11	21	12	21
Total	57	24	57	24
Coincidence rate	80.70%	87.50%	78.95%	87.50%

## Data Availability

All data generated and analyzed during this study are included in this published article.

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
