# Peer review of "On-Site and Visual Detection of the H5 Subtype Avian Influenza Virus Based on RT-RPA and CRISPR/Cas12a"

_viruses, 2024, doi:10.3390/v16050753_

Round 1

Reviewer 1 Report

Comments and Suggestions for Authors

Zhou et al develop a CRISPR/Cas-based assay for detecting H5 AIV strains with the focus on reducing cost and equipment required. Overall, this is an interesting study that could potentially be very interesting/impactful to many fields of virology. However, the writing, presentation of the data, and experimental details provided to the reader need to be dramatically improved. There are little-to-no experimental details given in the results section that allow the reader to understand how the assays are being performed and optimized. Furthermore, individual figure panels are not referenced or explained in the text. Figures are only referenced a single time by their number, figure panels are not called out nor explained in the text, samples are not labeled to allow for direct cross-comparison between assays, and often the panels are not labeled at all.

Major:

- More experimental details/rationale need to be provided in the results section for readers to adequately understand what is being assayed. For example, section 3.1 states that 6 crRNA were screened in two rounds of experiments for performance using an HA plasmid. But no information is given about what is happening in the assay, what the probes are, what the readout indicates, etc.

- In addition to the point above, the same information should be provided for the RT-RPA optimization. What primers are being optimized? Against the HA gene? If so from what strain? Also, the initial section says HA plasmid is being used, but the RT-RPA section states that the optimization was performed using an RNA template. Where is the RNA coming from? Was it purified from cells transfected with the HA plasmid?

- no explanation is given in the results section about what the lateral flow strips measure, what differences in lines indicates, and how that assay is performed.

- When the assay is deployed against H1-H16 subtypes from the different viruses, presumably the authors are using HA genes cloned into expression plasmids? Or are the nucleic acids being tested from live virus? Also, are these consensus sequences being tested from each strain or a specific isolate?

- Figure panels need to be referenced in the text and explained. For example, where are the explanations for Figures 4C and 4F in the text? What are these showing? What do they mean? Figure 5 has 4 separate graphs/images with no subpanels or descriptions provided.

- Positive and negative patient samples in Figure 6 need to be labeled so direct comparisons can be made between the fluorescence assay and lateral flow strips. Also, where are the data from the RT-qPCR assay? Are those the data in the first graph in Figure 6? This all needs to be labeled and described.

- Was the negative patient sample that screened positive in the RT-RPA/CRISPR assay confirmed that it should have been negative? In other words, did the authors sequence the nucleic acid to verify that it was actually a false-positive?

Minor:

- line 48 has some spurious commas

- sentence at lines 65-67 is confusing/possibly missing a word or two

Comments on the Quality of English Language

Minor grammatical changes need to be made

Author Response

Reviewer 1: Zhou et al develop a CRISPR/Cas-based assay for detecting H5 AIV strains with the focus on reducing cost and equipment required. Overall, this is an interesting study that could potentially be very interesting/impactful to many fields of virology. However, the writing, presentation of the data, and experimental details provided to the reader need to be dramatically improved. There are little-to-no experimental details given in the results section that allow the reader to understand how the assays are being performed and optimized. Furthermore, individual figure panels are not referenced or explained in the text. Figures are only referenced a single time by their number, figure panels are not called out nor explained in the text, samples are not labeled to allow for direct cross-comparison between assays, and often the panels are not labeled at all.

Response:

Thank you for your comments. We greatly appreciate your suggestions and have carefully revised our manuscript in response to each one. To aid in the reassessment process, we have marked all the changes in the revised manuscript with tracked changes. Additionally, we have corrected other minor errors found throughout the document. Below, we provide our responses to each reviewer's comment.

Major

Question 1: More experimental details/rationale need to be provided in the results section for readers to adequately understand what is being assayed. For example, section 3.1 states that 6 crRNA were screened in two rounds of experiments for performance using an HA plasmid. But no information is given about what is happening in the assay, what the probes are, what the readout indicates, etc.

Response: Thank you for the suggestions. We have revised the manuscript accordingly.

 In Section 3.1, for crRNA screening, we utilized the HA plasmid as a template for CRISPR/Cas12a detection and identified crRNAs with strong performance based on fluorescence levels. The probes used were fluorescent probes (6-FAM-TTATT-BHQ1); a higher fluorescence value indicates better crRNA effectiveness, as described in Section 2.5.

To enhance clarity for the readers, we have included additional information in the results section. Please refer to lines 171-180.

Question 2: In addition to the point above, the same information should be provided for the RT-RPA optimization. What primers are being optimized? Against the HA gene? If so from what strain? Also, the initial section says HA plasmid is being used, but the RT-RPA section states that the optimization was performed using an RNA template. Where is the RNA coming from? Was it purified from cells transfected with the HA plasmid?

Response:

What primers are being optimized? Against the HA gene? If so from what strain?

In this study, we optimized the primers for amplifying the HA gene of H5 subtype viruses. The goal was to establish a specific assay for H5 subtype avian influenza viruses (AIV), selecting the HA gene as the target due to its significant nucleotide sequence variation from other HA subtypes. We designed three forward and three reverse primers specifically for the H5 subtype AIV HA gene and identified the most efficient primer pairs through random combinations of these primers.

To clarify our methodology, we have added the following information: "We designed three pairs of primers to amplify the HA gene of H5 viruses after analyzing 988 HA sequences from Chinese H5 viruses." (Lines 192-194).

Also, the initial section says HA plasmid is being used, but the RT-RPA section states that the optimization was performed using an RNA template. Where is the RNA coming from? Was it purified from cells transfected with the HA plasmid?

For crRNA screening, we used the HA plasmid. However, for the RT-RPA section, optimization was carried out using RNA templates from virus A/Duck/GuangXi/1/2018 (H5N6), because RT-RPA involves both transcription and DNA fragment amplification. We chose virus RNA as it best mimics clinical samples. This information has been added accordingly (lines 195-196).

Question 3: no explanation is given in the results section about what the lateral flow strips measure, what differences in lines indicates, and how that assay is performed.

Response: Thanks for your professional comment.

no explanation is given in the results section about what the lateral flow strips measure,

The interpretation of the results of the lateral flow test strips were added in lines 212-216.

what differences in lines indicates, and how that assay is performed.

Additionally, we have included relevant information about the interpretation of the results from lateral flow test strips in the notes of Figure 1 (lines 81-85). Details on how the assay is performed are described in lines 139-144.

Question 4: When the assay is deployed against H1-H16 subtypes from the different viruses, presumably the authors are using HA genes cloned into expression plasmids? Or are the nucleic acids being tested from live virus? Also, are these consensus sequences being tested from each strain or a specific isolate?

Response: The templates used in these experiments were RNAs extracted from inactivated H1-H16 avian influenza viruses, preserved at the National Avian Influenza Reference Laboratory, Harbin Veterinary Research Institute of the Chinese Academy of Agricultural Sciences. These are described in lines 95-101.

To clarify, we have added information in lines 212-213 stating, “RNAs of H5 subtypes and other H1-H16 subtypes of AIV were extracted from inactivated AIVs and used as templates”.

The nucleotide sequences of these virulent strains were sequenced to obtain.

Question 5: Figure panels need to be referenced in the text and explained. For example, where are the explanations for Figures 4C and 4F in the text? What are these showing? What do they mean? Figure 5 has 4 separate graphs/images with no subpanels or descriptions provided.

Response: Thank you for your valuable comments.

The contents of Figure 4 are referenced and explained in Section 3.2 (lines 206-209). The contents of Figure 5 are referenced and explained in Section 3.3 (lines 226-227) and again in Section 3.4 (lines 230-232).

Subpanels and their corresponding explanations have also been added to the figure legends.

Question 6: Positive and negative patient samples in Figure 6 need to be labeled so direct comparisons can be made between the fluorescence assay and lateral flow strips. Also, where are the data from the RT-qPCR assay? Are those the data in the first graph in Figure 6? This all needs to be labeled and described.

Response: We sincerely appreciate the valuable comments.

We have labeled the positive and negative samples in Figure 6. Regarding the RT-qPCR data, it was not included in our previous study. We have revised the table 3-3 to clarify this (lines 252) and have also added a Supplementary Table (Table S1) to present the RT-qPCR data.

Question 7: Was the negative patient sample that screened positive in the RT-RPA/CRISPR assay confirmed that it should have been negative? In other words, did the authors sequence the nucleic acid to verify that it was actually a false-positive?

Response: The clinical samples used in this experiment were previously confirmed by virus isolation via egg inoculation. For any negative sample that screened positive in the RT-RPA/CRISPR assay, we did not verify whether it was a false positive by Sanger sequencing. In response to the reviewer’s question, we have addressed this point in the discussion section (lines 300-301).

Minor:

- line 48 has some spurious commas

Response: The commas have been corrected.

- sentence at lines 65-67 is confusing/possibly missing a word or two

Response: The sentence have been revised (Lines 66-67).

Comments on the Quality of English Language

Minor grammatical changes need to be made.

Response: Thank you for the comment. We have thoroughly corrected the grammar.

For data on RT-qPCRs, please see Table S1 in the Appendix.

Reviewer 2 Report

Comments and Suggestions for Authors

The manuscript presents the design the validation of a semi-point of care of assay. The design appears to be appropriate and the proposed application sound.  The manuscript defines the expected applications outside of the diagnostic lab.  Although these in field results would likely need to be confirmed, this technology represents another tool aimed at early detection of avian influenza virus, mainly viruses belonging to the H5 subtype. I have no issue with this manuscript at all. 

Line 30: Change Avian Influenza Viruses to Avian influenza viruses.

  Figure 1. Change "naked eyes" to something like "Visual Inspection" or "Plain Sight" 

Author Response

On behalf of all contributing authors, I would like to express our sincere appreciation for your constructive comments on our manuscript. Your feedback has been invaluable in improving our article. In response to your comments, we have revised the manuscript to enhance its rigor. In this revised version, all changes to our manuscript are highlighted.

Reviewer 3 Report

Comments and Suggestions for Authors

Thank you for the opportunity to review the original research article entitled "On-site and Visual Detection of H5 Subtype Avian Influenza Virus Based on RT-RPA and CRISPR/Cas12a" by Zhou et al.

The work presents promising results from a contemporary and accurate diagnostic approach study for monitoring and control of one the most significant zoonotic viral pathogens - H5N1 avian influenza virus. The continuous spread and emergence of novel viral variants necessitates development and introduction of rapid, specific, easy, cheep and flexible technology for testing and disease limitation. CRISPR-Cas system has been intensively explored in the latest years as a reliable diagnostic tool for various pathogenic agents with the trend to be advantageous to other molecular methods (qPCR) in terms of specificity, detection limits and cost.

The presented research confirms this trend and is in concordance with similar studies on diagnostics of other viral pathogens in clinical samples. In this view, I find it essential additional citations, comparisons with other authors' results and more detailed analysis on particular data to be provided in the Discussion section of the paper to support the conclusions made. Also, I was surprised not seeing milestone papers in the field to support the background and rationale - Kaminski et al., 2021; Park et al., 2021, Lou et al., 2022; Tripathi et al., 2023, Chen et al., 2023; Yang et al., 2023, etc. It my opinion, the Introduction and Discussion sections, as well as the references could be extended and enriched a little.

There is scarce information about samples origin -  species, place(s) of collection, storage conditions and period of preservation. That could be crucial for the test performance and reproducibility of results.

There is a typo (seems to me) in Line 198 - Figure 1B should be Figure 2B, actually.

Despite the above mentioned remarks, the work is designed, implemented and presented in a qualitative manner, methods and results come logically in order, tight style, clear, correct and comprehensive language, figures are easy to understand. I would suggest the article for publication with minor revision.
